# Non-linear links between human capital, educational inequality and income inequality, evidence from China

**Mo Xu[1], Shifeng Chen[2], Jian Chen[3], Taiming Zhang[ID][4]***

**1** Faculty of Finance, City University of Macau, Macau, China, **2** Postal Saving Bank of China, Zhejiang, China, **3** Faculty of Social and Historical Sciences, College London, London, United Kingdom, **4** Finance Department, Business School, University of Edinburgh, Edinburgh, United Kingdom

* taimingzhang1990@outlook.com

**Data Availability Statement:** All relevant data are available at https://databank.worldbank.org/reports.aspx?source=World-Development-Indicators.

**Funding:** The author(s) received no specific funding for this work.

## Abstract

This study aims to reveal short-run and long-run asymmetries among human capital, educational inequality, and income inequality in China over the period 1975–2020 using a nonlinear autoregressive distributed lag (NARDL) model. The estimated long-run asymmetry parameters reflect that positive shocks to secondary education (SSE) and higher education (HE) are negatively correlated with income Gini coefficient. The adverse shocks of secondary education (SSE) and higher education (HE) stimulate the Gini coefficient of income, but the effect of secondary education (SSE) on the Gini coefficient of income is not significant, while that of higher education (HE) is significant. The results also highlight that, in the long run, there is a significant asymptotic effect of the education Gini coefficient (educational inequality) and economic growth on the income Gini coefficient (income inequality). However, physical capital stock has a significant adverse effect on income inequality in the long run. Higher education significantly promotes educational inequality, while the square of higher education significantly reduces educational inequality, thus verifying the inverted U-shaped Kuznets curve hypothesis between higher education and educational inequality. Strategically, this study suggests higher education as a powerful tool for mitigating income inequality by emphasizing educational equity.

## 1. Introduction

Global policymakers now increasingly concerned about rising income inequality [1]. Over the past decade, income inequality has been observed to parallel economic growth in many countries [2]. High income inequality has serious implications for sustainable economic growth, leading to financial and economic uncertainty that inhibits investment [3]. Income inequality affects developing economies much more severely, with low incomes and large income gaps leading to poverty, low levels of education, inefficient markets and malnutrition [4].

The attention of researchers and academics has been drawn to factors attributable to rising income inequality. As one of the main factors affecting the income distribution of social groups, human capital has been identified by researchers [5, 6]. The set of factors embodied in

**Competing interests:** The authors have declared that no competing interests exist.

a person, such as skills training, health and education, is known as human capital. The enhancement of all these dynamics generates human capital accumulation [7]. According to the economic literature, human capital (measured by the level of education attained) is a major determinant of personal income, since it has been observed that the less educated have a lower chance of earning a high income than the more educated. Human capital generation in socially underdeveloped areas may be constrained by income inequality, which may lead to human capital inequality, again undermining fair income distribution, forming a vicious circle [8]. In addition, developing countries can improve economic performance by accumulating human capital through increased spending on education, which can help reduce inequality of opportunity and break the intergenerational transmission of poverty, leading to a more equitable income distribution [6]. Parents believe that investing in their children's education is an important way to increase their children's future income. An effective tool for reducing educational inequality and thus income inequality is to increase government spending on education. In recent years, as educational attainment has increased in many countries, income inequality has widened and educational inequality has shrunk. As the researchers predicted, income inequality shrank as average educational attainment and educational equality rose.

At present, the problem of income inequality is prominent. In some countries, the gap between the rich and the poor and the collapse of the middle class have exacerbated social polarization, political polarization and populationism, which has brought profound lessons to the world [9]. China must resolutely prevent polarization, promote common prosperity, and realize a harmonious society [10]. Now, with the shared-prosperity agenda put on hold and the economy reeling from endless COVID-19 lockdowns, China's wealth gap could widen. In the coming years, income inequality could become a major challenge for China's development and a major test for Xi Jinping and the party [11]. Published estimates of the Gini coefficient for 2022 show a high level of inequality in China at 0.63 [12]. In terms of overall wealth, China has a wider gap between rich and poor, not just income but all assets such as stocks, bonds and real estate. Taking these factors into account, the World Inequality Lab found that the richest 10 percent of China's population own nearly 70 percent of total household wealth [13]. Research carried out at the onset of the pandemic has shown that covid restrictions are eating into rural wages. According to a 2020 rural survey by Stanford University researchers, nearly three-quarters of respondents said people who typically work in the field have been unable to do so because of the pandemic [14]. More than 90% said covid controls had reduced their income, and such restrictions on movement have returned in the past six months. China's high income inequality statistics mean that lower-income segments of society have limited access to education and employment opportunities, again leading to a widening income gap. The high levels of income and wealth inequality in recent decades call for active public policies to minimize the impact of inequality on China's economic development. This study predicts that low educational attainment and widening educational gaps are important factors leading to income inequality in China. In 2020, 37% of China's population aged 25–64 had at least a high school education, compared with the OECD average of 83%. Among people aged 25–64 in China, short-cycle higher education is the most common form of higher education, accounting for 10% of the total population, followed by undergraduates at 8%, and masters and doctors combined at 1%. This statistic differs from the OECD average, where a bachelor's degree is the most common at 19%, followed by a master's degree at 14%, and a short-term higher education qualification at 7% [15]. Scholars argue that expanding education will, in due course, increase enrollment; thus, it will reduce the gap in educational opportunity, as children from disadvantaged backgrounds will be enrolled at a much higher rate compared to upper-class ones. The plan to expand education is seen as a shift from elite to mass education. Clearly, education level and distribution are key subsets of income distribution [16].

Research tends to highlight educational attainment and education inequality are the main factors influencing the level of income inequality. Despite widespread public and policy-maker awareness and interest in the importance of education on income distribution, the relationship between educational attainment and its distribution among income-equal groups specifically in the Chines context has not always been explored theoretically and empirically [17]. Thus, this paper empirically explores the important factors of income inequality in China over the past four and a half decades, and clarifies the relationship between educational acquisition and distribution and income distribution. Furthermore, the aim of exploring the link between educational attainment and income inequality using time-series data of the Chinese economy is based on the fact that China is not only the fastest growing major economy, but also that income inequality is expanding rapidly. Besides, China's high income inequality is also the main reason for the increasing inequality of world income distribution. In addition, this study also reveals the authenticity of the Kuznets curve between educational attainment and educational inequality in the Chinese context. The problems of heterogeneity and data comparability often encountered in cross-country studies can be alleviated by choosing a single-country study in China. Moreover, this study employs a nonlinear autoregressive distributed lag (NARDL) bounds test to extend the econometric analysis to account for nonlinearities in the relationship. Including this asymmetry is crucial because gradual or adverse changes in one factor will not have the same effect on the other. Also, different channels of positive and negative shocks can be identified with the aid of asymmetric relationships.

## 2. Literature review

[18] pointed out that the impact of educational expansion on income distribution is ambiguous. They show that educational expansion has two countervailing effects on income distribution: One is a "constitutional effect," whereby early wage inequality is stimulated by educational expansion, leading to an increase in the proportion of well-educated workers. The other is the "wage compression effect," whereby when education expansion creates a surplus of educated labor, the premium for educated workers eventually falls, reducing wage inequality. When exploring the relationship between education and income inequality, empirical literature based on cross-country data often presents conflicting results. [19] reveals the role of information and communication technology (ICT) in curbing the impact of education and lifelong learning on income inequality and economic growth from 2004 to 2014 in a sample of 48 African countries. The results show that mobile phones and the Internet each interact with primary education to reduce income inequality, while all ICT indicators interact with secondary education to adversely affect income inequality. More recently, [20] investigated the relationship between income inequality, educational attainment, and CO2 emissions by using a Dynamic Common Correlation Effect (DCCE) estimator for 64 countries from 1990 to 2016. The results of the analysis show that educational attainment and $CO_2$ emissions are negatively correlated with income inequality in selected countries. [5] also documented that educational expansion is a major factor in reducing income inequality using cross-country data for East Asian economies from 1980 to 2015. In contrast, more recently, [21] found that increases in human capital lead to lower poverty levels, however, human capital is positively associated with income inequality, indicating unequal economic opportunities and unequal educational systems. [22] applied panel cointegration and fully modified OLS to develop a quadratic relationship between education and income inequality in developing Asia over the period 1960–2015. The analysis results show that primary school, secondary school and university enrollment promote income inequality, but the impact of education on income inequality becomes negative after reaching a certain threshold. [23] examine the impact of mean years of schooling

and educational inequality on income inequality in South Asian countries from 1980 to 2010 using a fixed effects model (FEM) and a random effects model (REM). The findings show that average years of schooling and educational inequality significantly exacerbate income inequality in South Asian countries.

The empirical literature on the link between educational expansion and income inequality also shows ambiguous results for individual countries. [24] take a closer look at the link between education and income inequality using a balanced panel dataset from Greece over the period 1994–2012. The results show that education level has an adverse effect on the formation of income inequality in Greece. [6] also provide empirical evidence on the short- and long-run asymmetries between human capital and income inequality in India from 1970 to 2016 using the NARDL bound testing approach. The findings highlight that expanding education is a major factor in reducing the high income inequality that prevails in India. Another study by [25] highlights that human capital (educational attainment) has become a key determinant of reducing income inequality in Portugal. Likewise, [26] provide evidence using the ARDL approach that higher education has a significant adverse effect on income inequality in Pakistan from 1973 to 2012. Another study by [27] attempts to provide evidence for the significant adverse effects of educational expansion on income inequality in China using survey data and the FFL decomposition method. [28] used the ARDL approach and data from 1969 to 2007 to conclude that increases in human capital can reduce income inequality and thus make income distribution more equitable in Iran. In contrast, [29] predict that, assuming countries are heterogeneous and interdependent cross-sections, human capital is the strongest determinant of inequality, which, as theory predicts, will exacerbate inequality. Conversely, [30] conclude that in Israel the proportion of households with higher education has increased, as has the proportion of households with full-time dual-earners. There has also been an increase in the proportion of dual-blessed households: those with higher education in Israel and full-time dual-earner incomes, all of which contribute to a rise in income inequality. Similarly, [31] also conclude that educational expansion in Turkey or technological development based on education and research stimulates income inequality.

[32] acknowledge that sociological theories of social closure reflect that inequality in educational attainment is more important in predicting income inequality than skill inequality. Using education policy as an instrument, educational inequality appears to be a stronger predictor of income inequality than skill acquisition inequality. [33] explore a broad, positive, statistically significant and stable relationship between educational inequality and income inequality, especially in emerging and developing economies. [34] also reveal a significant progressive effect of educational inequality as a proxy for human capital inequality on income inequality. [23] documented that unequal educational distribution of boys and girls in tertiary education reduces income inequality while increasing income inequality in primary school. Using educational attainment as a measure of human capital, [5] examined the impact of educational attainment on income inequality in a panel of 95 economies. Findings suggest that expanding education leads to educational equality, which in turn leads to more equal income distribution.

Existing literature using data on education and income inequality in individual countries is also broadly consistent with findings from cross-country studies. Overall, the effect of educational expansion on income distribution is ambiguous, while the effect of educational inequality on income distribution is unequal. In addition, we have not found any research on the effectiveness of the Kuznets curve in the relationship between educational attainment and educational inequality in the Chinese context.

## 3. Model building, variable data measurements and sources, and methods

### 3.1 Theoretical framework

The analysis of the relationship between education level and income inequality can be based on the theoretical framework of the traditional human capital model proposed by [35, 36]. The underlying concepts of the theoretical framework can be used in the current study to construct our empirical model. In the standard theoretical framework proposed by [35, 36], the educational level and distribution of the population are the main determinants of income distribution. Thus, the model clearly shows that disparities in income distribution can be uncovered through the demand and supply of educated workers.

Considering factors such as Y for personal income and S for years of education used to measure human capital by [37, 38] can be approximated as:

$$\log Y_s = \log Y_0 + \sum_{j=0}^{s} log(1 + r_j) + \varepsilon \qquad (1)$$

where, Ys denotes the income level of individuals with an educational level of S, $Y_0$ represents the income level of individuals without formal education, rj is the return rate in the jth year, and $\varepsilon$ indicates the random error term. [6] also follow the variable relationships mentioned in the above model in the context of India; [21] in the context of Eastern Cape Province, South Africa. This function can be approximated as:

$$\log Y_s = \log Y_0 + rS + \varepsilon \qquad (2)$$

The above function with logging can be converted to

$$\text{Var}(\log Y_s) = \bar{r}^2 \text{Var}(S) + \bar{S}^2 \text{Var}(r) + 2\bar{r}\bar{S}\,\text{Cov}(r, S) + \text{Var}(\varepsilon) \qquad (3)$$

According to [5], controlling for other factors (ceteris paribus), educational expansion reduces educational inequality, which in turn reduces income inequality. However, [6] argue that this relationship becomes blurred when the value of r (return to education) is coagulated with educational inequality. Furthermore, when r and S self-regulate each other, an increase in S (educational expansion) will lead to higher income inequality.

### 3.2 Empirical model development

Based on the basic concepts of the above theoretical framework, this study constructs an empirical model to analyze the impact of education level on income inequality. This study uses annual time series data from 1975 to 2020 to explore the empirical relationship between secondary education attainment and higher education attainment represented by human capital, educational inequality, economic growth, physical capital, and income inequality. The choice of data period is based on the availability of data related to all variables, with higher sample data giving more reliable results than lower data selection.

This relationship can be expressed by the following econometric models.

$$\ln G_{I,t} = \beta_0 + \beta_1 lnSSE_t + \beta_2 \ln HE_t + \beta_3 \ln G_{E,t} + \beta_4 \ln GDP_t + \beta_5 \ln K_t + \mu \qquad (4)$$

$$\ln G_{E,t} = \beta_0 + \beta_1 lnSSE_t + \beta_2 \ln SSE_t^2 + \beta_3 \ln HE_t + \beta_4 \ln HE_t^2 + \mu \qquad (5)$$

where, $\beta_1$, $\beta_2$, $\beta_3$, $\beta_4$ and $\beta_5$ symbolize estimated coefficients, t signify time, ln is natural logarithm, $\mu$ indicate stochastic error term. $G_{I,t}$ and $G_{E,t}$ represent Gini coefficient (measure of income inequality) and Gini coeffeicient (measure of education inequality) respectively, $SSE_t$,

**Table 1. Description and measurement of variables and data sources.**

| Variables | Description | Measurment | Sources |
|---|---|---|---|
| GDP | Gross Domestic Product | Constant 2015 US$ | WDI, World Bank (2022) |
| $.G_I$ | Income Gini coeffecient | Net income Gini index | World income inequality database (WIID) |
| $G_E$ | Education Gini coeffecient | Education Gini index | WDI, World Bank (2022) |
| SSE | Secondary School Education attainment | Barro-Lee: Average years of secondary schooling, age 15+ | WDI, World Bank (2022) |
| HE | Higher education attainment | Barro-Lee: Average years of higher schooling, age 25+ | WDI, World Bank (2022) |
| K | Gross fixed capital formation | Constant 2015 US$ | WDI, World Bank (2022) |

$HE_t$ are the secondary school attainment and higher education attainment respectively, $SSE_t^2$ and $HE_t^2$ are the squared of secondary school attainment and higher education attainment respectively, $GDP_t$ and $K_t$ denote gross domestic product and capital stock respectively.

Variable measures, descriptions, and data sources are highlighted in the (Table 1) below, and reflect the income Gini coefficient as an indicator of income inequality, as measured by the Net Income Gini Index provided by the World Income Inequality Database (WIID). As a proxy for educational inequality, the Education Gini Coefficient is measured by the Education Gini Index, available from the World Bank's World Development Indicators (WDI). Secondary education attainment (Barro-Lee: Average years of secondary schooling, age 15+) and higher education attainment (Barro-Lee: Average years of higher schooling, age 25+) data are available from WDI, World Bank. Gross fixed capital formation (Constant 2015 US$) and gross domestic product (GDP) (Constant 2015 US$) data are available from the World Bank's public website World Development Indicators (WDI).

### 3.3 Methodology

**3.3.1 Non-linear autoregressive distributive lag (NARDL).** A relatively new asymmetric or nonlinear ARDL method proposed by [39] used in the current study to detect long-term and short-term asymmetries between variables. [40, 41] suggest that the NARDL model adapted for the current study outperforms traditional ARDL techniques in examining small-sample cointegration. The current literature identifies the use of various estimation techniques to detect the impact of human capital development on income inequality, such as the traditional linear ARDL model of [42], the two-stage least squares (2SLS) method of [43]. But no studies have scrutinized the asymmetric link between human capital development and income inequality. This is thus at odds with [44] argument that wage adjustments can be sticky and asymmetric, which we argue affects asymmetry in the distribution of income. Hence, this study aims to fill the gap by exploring the asymmetric impact of human capital development on income inequality in the Chinese economy. This approach is employed in various studies to examine whether expansion or contraction of regressors affects the regressand differently. Considering [39], asymmetric cointegration regression,

$$Yt = \beta^+ xt^+ + \beta^- xt^- + \mu_t \tag{6}$$

where β+ and β− reflect the asymptotic and inverse long-term parameters, and xt is a k × 1 vector of regressors decomposed as:

$$xt = x_o + xt^+ + xt^- \tag{7}$$

Following Eqs (6) and (7), the baseline model (i.e., Eq (4)) is transformed into an asymmetric equation by substituting $SSE_t$ and $HE_t$ for partial positive and negative sum decomposition.

$$\ln G_{I,t} = \beta_0 + \beta_1 lnSSE_t^+ + \beta_2 lnSSE_t^- + \beta_3 \ln HE_t^+ + \beta_4 lnHE_t^- + \beta_5 \ln G_{E,t} + \beta_6 \ln GDP_t + \beta_7 \ln K_t$$
$$+ \mu \tag{8}$$

In Eq (8), the motion in $SSE_t$ and $HE_t$ decomposed into its progressive and regressive parts, that is, $SSE_t = SSE_t^+ + SSE_t^-$, $HE_t = HE_t^+ + HE_t^-$, where the positive and negative signs represent the increase and decrease of each $SSE_t$ and $HE_t$ respectively.

Eqs (9), (10), (11) and (12) below can express the partial sums of the increasing and decreasing changes for each of $SSE_t$ and $HE_t$.

$$SSE_t^+ = \sum_{i=1}^{t} \Delta SSE_t^+ = \sum_{i=1}^{t} \max(\Delta SSE_t, \, 0) \tag{9}$$

$$SSE_t^- = \sum_{i=1}^{t} \Delta SSE_t^- = \sum_{i=1}^{t} \max(\Delta SSE_t, \, 0) \tag{10}$$

$$HE_t^+ = \sum_{i=1}^{t} \Delta HE_t^+ = \sum_{i=1}^{t} \max(\Delta HE_t, \, 0) \tag{11}$$

$$HE_t^- = \sum_{i=1}^{t} \Delta HE_t^- = \sum_{i=1}^{t} \max(\Delta HE_t, \, 0) \tag{12}$$

Following [39], substituting the positive and negative sums of each $SSE_t$ and $HE_t$ respectively, we obtain the following asymmetric cointegration equation.

$$\Delta G_{I,t} = \beta_0 + \Psi G_{I,t-1} + \Psi^+ SSE_{t-1}^+ + \Psi^- SSE_{t-1}^- + \Psi^+ HE_{t-1}^+ + \Psi^- HE_{t-1}^- + \Psi G_{E,t-1} + \Psi GDP_{t-1}$$
$$+ \Psi K_{t-1} + \sum_{i=1}^{q} \beta_i \Delta G_{I,t-1} + \sum_{i=1}^{q} \rho_i^+ SSE_{t-1}^+ + \sum_{i=1}^{q} \rho_i^- SSE_{t-1}^- + \sum_{i=1}^{q} \rho_i^+ HE_{t-1}^+$$
$$+ \sum_{i=1}^{q} \rho_i^- SSE_{t-1}^- + \sum_{i=1}^{q} \rho_i \Delta G_{E,t-1} + \sum_{i=1}^{q} \rho_i \Delta GDP_{t-1} + \sum_{i=1}^{q} \rho_i \Delta K_{t-1}$$
$$+ \mu_t \tag{13}$$

The asymmetric ARDL cointegration method consists of several steps, contained in Eq (13) above. First, after evaluating the null hypothesis $H_o = \theta^+ = \theta^-$ against its alternative $H_1 = \theta^+ \neq \theta^-$, a Wald test can be used to estimate the long-run nonlinear effect. The null hypothesis rejection describes the existence of an asymmetric or non-linear effect of secondary and tertiary education on income inequality. The $\theta+$ and $\theta-$ reflect long-run positive and negative changes while $v_i^+ = v_i^-$ or $\sum_{i=1}^{p} v_i^+ = \sum_{i=1}^{p} v_i^-$ implies the short-run asymmetric effects of progressive and inverse variations in each $SSE_t$ and $HE_t$.

The asymmetric dynamic multiplier effect of $SSE_{t-1}^+$, $SSE_{t-1}^-$, $HE_{t-1}^+$ and $HE_{t-1}^+$ unit changes can be further estimated, defined as follows:

$$m_p^+ = \sum\nolimits_{k=0}^{p} \frac{\lambda G_{I,t+k}}{\lambda SSE_t^+} \tag{14}$$

$$m_p^- = \sum\nolimits_{k=0}^{p} \frac{\lambda G_{I,t+k}}{\lambda SSE_t^-} \qquad P = 0,1,2,\ldots. \tag{15}$$

$$m_p^+ = \sum\nolimits_{k=0}^{p} \frac{\lambda G_{I,t+k}}{\lambda HE_t^+} \tag{16}$$

$$m_p^- = \sum\nolimits_{k=0}^{p} \frac{\lambda G_{I,t+k}}{\lambda SSE_t^-} \qquad P = 0,1,2,\ldots. \tag{17}$$

Note: $m \to \infty$, $m_p^+ \to \beta^+$ and $m_p^- \to \beta^-$, where $\beta^+$ and $\beta-$ denote the long-run asymmetry coefficients respectively, as defined above.

The recursive cumulative sum of residuals (CUSUM) and recursive cumulative sum of squared residuals (CUSUMSQ), developed by [45], will be used in the current study to test the stability of the model.

# 4. Results and interpretation

First, Augmented Dickey-Fuller (ADF) [46] and Kwiatkowski–Phillips–Schmidt–Shin (KPSS) [47] are two different unit root tests used in the current study to examine the order of integration of all variables in the model for analysis. Describing the level of stationarity of each variable by a unit root test is an important step to consider before proceeding to explore cointegration among variables. Such as, the ARDL technique cannot be used if any variable integrates to order I(2) [46, 47]. Table 2 highlights the results of ADF and KPSS tests on the level of stationarity, reflecting that all variables are first-order integrated, i.e., I(1), except for GDP and capital stock, which are level-integrated, i.e., I(0), without variable integration to the second order. This result clearly confirms the suitability of the ARDL technique for further analysis.

In the presence of structural breaks, the unit root tests of [46, 47] may be biased because these tests do not accommodate the presence of structural breaks in the series. however, the [48] unit root test can address this issue because the test includes an unknown single and two

**Table 2. Unit root test results without structural break.**

| Variables | ADF | | KPSS | |
| --- | --- | --- | --- | --- |
| | Level | First difference | Level | First Difference |
| $\ln G_{I,t}$ | -2.68 | -3.14*** | 0.33 | 0.71*** |
| $\ln SSE_t$ | -3.13 | -4.33*** | 0.82 | 0.35*** |
| $\ln HE_t$ | -4.38 | -5.91*** | 0.34 | 0.23*** |
| $\ln G_{E,t}$ | -2.23 | -4.03*** | 0.68 | 0.39*** |
| $\ln GDP_t$ | -4.61*** | -6.91*** | 0.45*** | 0.33*** |
| $\ln K_t$ | -3.57** | -3.93*** | 0.908* | 0.65*** |

Note

*, **, *** indicate statistical significance levels of 10%, 5%, and 1%, respectively.

**Table 3. Clemente-Montanes-Reyes unit root test results with unknown structural breaks.**

| Variables | Innovative outliers | | | | Additive outliers | | | |
|---|---|---|---|---|---|---|---|---|
| | $T_{a1}$ | $T_{a2}$ | Statistics | K | $T_{a1}$ | $T_{a2}$ | Statistics | K |
| $LnG_{I,t}$ | 1978 | - | -3.01 | 2 | 1983 | - | -6.03*** | 3 |
| | 1978 | 1989 | -3.24 | 3 | 1983 | 1989 | -5.31** | 2 |
| $lnSSE_t$ | 1989 | - | -4.38 | 1 | 2005 | - | -8.42*** | 4 |
| | 1989 | 1995 | -2.08 | 2 | 2005 | 1999 | -7.39*** | 3 |
| $lnHE_t$ | 2001 | - | -3.18 | 3 | 2003 | - | 12.14*** | 4 |
| | 2001 | 2005 | -3.86 | 2 | 2003 | 2007 | -9.28*** | 3 |
| $lnG_{E,t}$ | 1983 | - | -1.37 | 3 | 2004 | - | -9.08*** | 3 |
| | 1983 | 2003 | -1.54 | 3 | 2004 | 2009 | -4.21** | 2 |
| $lnGDP_t$ | 1988 | - | -3.52 | 2 | 2005 | - | 15.85*** | 2 |
| | 1988 | 2007 | -2.85 | 3 | 2005 | 2011 | 11.32*** | 3 |
| $lnK_t$ | 1985 | - | -2.97 | 2 | 1975 | - | -9.68*** | 3 |
| | 1985 | 2002 | -4.13 | 3 | 1975 | 2004 | 11.32*** | 4 |

Note

*, **, *** indicate statistical significance levels of 10%, 5%, and 1%, respectively.

structural break in the variables data. In addition, the test adjusted for structural breaks in the trend function in the null and alternative hypotheses of unknown dates. The results highlighted in Table 3 reflect that income inequality, secondary education, tertiary education, educational inequality, economic growth, and capital formation are stationary at first difference in the presence of single and double unknown structural breaks in the series.

Next, bounds test analysis can be performed to examine the value of the F statistic to explore the presence or absence of long-run relationships. The mixed order of integrals I(0) and I(1) demonstrates the effectiveness of using the bounds test method, and the highlighted results in Table 4 show the values of the F statistic as 4.778 and 5.893, exceeding the upper critical value with a significance level of 1%. Thus, the results clearly show that there are stable long-term relationships among the variables included in the two models.

The long-term parameters of the NARDL model are shown in Table 5, and asymmetric effects can be observed from the positive and negative partial sums of SSE and HE, namely, $SSE^+$ and $SSE^-$, $HE^+$ and $HE^-$ [49]. In the analysis results, the positive and negative shocks of SSE and HE both show a highly significant state with opposite signs, indicating that the increase or decrease of SSE and HE have different impacts on income inequality. Positive shocks to SSE and HE (human capital) are negatively correlated with the income Gini coefficient. This means expanding education to promote employment opportunities, thereby

**Table 4. Result of bound cointegration results.**

| | Model 1 | Model 2 |
|---|---|---|
| F-statistics | 4.778*** | 5.893*** |
| Lower-upper Bound (1%) | 3.39–4.56 | 3.13–4.41 |
| Lower-upper Bound (5%) | 2.87–3.75 | 2.42–3.56 |
| Lower-upper Bound (10%) | 2.21–3.32 | 2.08–3.25 |
| K | 5 | 4 |

Note

*, **, *** indicate statistical significance levels of 10%, 5%, and 1%, respectively.

**Table 5. Asymmetric ARDL model long-term coefficient elasticities results.**

| Variables | Model 1 | Model 2 |
|---|---|---|
| $\ln SSE_{t-1}^{+}$ | -0.731** (-0.024) | -0.192** (-0.031) |
| $\ln SSE_{t-1}^{-}$ | 0.372 (0.271) | --------------------- |
| $\ln HE_{t-1}^{+}$ | -0.512*** (-0.003) | 0.381***(0.002) |
| $\ln HE_{t-1}^{-}$ | 0.412*** (0.004) | ------------------ |
| $\ln G_{E,t-1}$ | 0.813*** (0.001) | ------------------ |
| $\ln GDP_{t-1}$ | 0.618*** (0.007) | --------------------- |
| $\ln K_{t-1}$ | -0.584*** (-0.008) | ---------- |
| $\ln SSE_{t-1}^{2}$ | --------------------- | 0.301**(0.031) |
| $\ln HE_{t-1}^{2}$ | -------------------- | -0.131***(-0.005) |
| Constant | −11.912***(0.001) | −9.851***(0.003) |
| $R^2$ | 0.99 | 0.78 |
| Adj $R^2$ | 0.63 | 0.74 |
| F-statistic | 922.112 | 983.603 |

Note

*, **, *** represent the statistical significance levels of 10%, 5%, and 1%, respectively, and the values in brackets are the probabilities of each coefficient.

reducing income inequality and achieving a more equitable income distribution. It can be seen from the analysis results that for every 1 percentage point increase in SSE and HE, income inequality can be significantly reduced by 0.731 percentage points and 0.512 percentage points, respectively. These findings fit well with theoretical concepts based on human capital models and confirm that educational expansion narrows the income inequality gap. These results support the findings of [5, 19–21]. The adverse shocks of SSE and HE stimulate the Gini coefficient of income, but the effect of SSE on the Gini coefficient of income is not significant, while that of HE is significant. A 1 percentage point worsening in SSE and HE yields an insignificant 0.372% and a significant 0.412% increase in the income Gini coefficient, respectively. In our analysis, both HE[+] and HE[-] appear to be highly significant and of opposite sign, suggesting that both increases and decreases in HE have adverse effects on income inequality.

This study also considered the relationship between changes in the income Gini coefficient (income inequality) and education Gini coefficient (educational inequality), showing that there is a positive relationship between changes in income inequality and educational inequality. The analysis shows that every 1 percentage point increase in the education Gini coefficient can significantly stimulate a 0.813 percentage point increase in the income Gini coefficient. This result is in good agreement with [5, 23, 32–34]. The impact of economic growth on the Gini coefficient of income is significantly positive in the long run, thus exacerbating income inequality. In the long run, a 1 percent increase in GDP can significantly increase income inequality by 0.618 percent, implying that increases in production levels do not contribute to a fair share of income. This clearly reflects that in China, the benefits of increased production are in the hands of very few. Moreover, consistent with [50, 51], in the long run, a positive change in the capital stock reduces the income Gini coefficient, reflecting a progressive effect on fair income distribution. Higher investment in the physical capital stock generates employment and ultimately leads to a more equitable distribution of income. In the long run, every 1 percentage point increase in capital stock can significantly reduce the income Gini coefficient (income inequality) by 0.584 percentage points.

The second model based on the Gini coefficient of education (educational inequality) shows that secondary education has a significant negative impact on the Gini coefficient of

education, while the square of secondary education has a significant positive impact on the Gini coefficient of education. This relationship between educational inequality and secondary educational attainment clearly validates the U-shaped hypothesis of the Kuznets curve. The coefficient of higher education is significantly positive, while the coefficient of the square of higher education is significantly negative, reflecting that higher education significantly promotes educational inequality, while the square of higher education significantly reduces educational inequality. This means that higher education contributes to educational inequality in the initial stage, and after reaching a certain threshold, the impact of higher education on educational inequality becomes negative. This relationship between higher education and educational inequality clearly verifies the inverted U-shaped hypothesis of the Kuznets curve in China.

In the short run, the empirical results highlighted in Table 6 are very similar to the long-run results for educational attainment (human capital). The results show that positive shocks to secondary education are negatively correlated with income inequality in the short run, suggesting that higher levels of secondary education lead to a more equitable income distribution in China. However, the positive shock of higher education has no significant impact on China's income distribution in the short run. Negative shocks to both levels of education do not have a significant impact on income inequality in the short run. In addition, physical capital stock, economic growth, and education Gini coefficients are significantly and positively correlated with income inequality in the short run.

In the second model, the short-term effect of education level on the Gini coefficient of education (educational inequality) is quite different from the long-term effect. Both secondary education (SSE) and the square of secondary education ($SSE^2$) have a significant adverse effect on the education Gini coefficient. However, both higher education and the square of higher education contribute significantly to the education Gini coefficient (educational inequality) in the short run. This means that secondary education can, in the short term, play a more critical role in closing the education gap than tertiary education.

The error correction term ($ECT_{t-1}$) is the speed adjustment coefficient, which is statistically significant and negative, reflecting that short-term shocks can be balanced in the long run. The

**Table 6. Asymmetric ARDL model short-term coefficient elasticities results.**

| Variables | Model 1 | Model 2 |
|---|---|---|
| $\ln SSE^+_{t-1}$ | -0.191*** (-0.004) | -0.208*** (-0.001) |
| $\ln SSE^-_{t-1}$ | 0.162 (0.351) | -------------------- |
| $\ln HE^+_{t-1}$ | -0.218 (-0.023) | 0.251*** (0.006) |
| $\ln HE^-_{t-1}$ | 0.422 (0.134) | ------------------ |
| $\ln G_{E,t-1}$ | 0.263** (0.021) | ------------------ |
| $\ln GDP_{t-1}$ | 0.308*** (0.002) | -------------------- |
| $\ln K_{t-1}$ | 0.221** (0.018) | ---------- |
| $\ln SSE^2_{t-1}$ | -------------------- | -0.481*** (-0.001) |
| $\ln HE^2_{t-1}$ | ------------------- | 0.183** (0.025) |
| Constant | −14.102***(0.005) | −11.041*** (0.001) |
| $R^2$ | 0.95 | 0.71 |
| Adj $R^2$ | 0.65 | 0.54 |
| F-statistic | 955.109 | 906.323 |

Note

*, **, *** represent the statistical significance levels of 10%, 5%, and 1%, respectively, and the values in brackets are the probabilities of each coefficient.

**Table 7. Results of sensitivity tests.**

| Test | Model 1 | Model 2 |
|------|---------|---------|
| RAMSEY | 1.984 (−0.862) | 1.108 (−0.611) |
| JB | 5.368 (−0.527) | 8.632 (−0.634) |
| ARCH | 1.958(−0.853) | 1.763 (−0.436) |
| RESET | 4.709 (−0.879) | 8.164 (−0.742) |
| LM | 1.746 (−0.825) | 0.706 (0.467) |

**Note:** Parentheses enclose the p-values

coefficients of $ECT_{t-1}$ are -0.73 and -0.86, respectively, suggesting that the short-term gap can be adjusted to the long-term balance in the range of 73%-86%.

RAMSEY, RESET, JB, LM, and ARCH are diagnostic tests that can be used to check for autocorrelation and heteroscedasticity problems, and the results in Table 7 clearly highlight the absence of autocorrelation and heteroscedasticity for the selected variables in the model. The variables included in each model had no serial correlation, as evidenced by the derived non-significance of the F-statistic demonstrated by the Breusch-Godfrey LM test.

The CUSUM and CUSUMSQ tests highlighted in (Figs 1 and 2) can be used to check the stability of the model, which clearly shows that the graph is within the critical range at the 5% significance level, thus validating the stability of the two estimated models.

Finally, dynamic multipliers can be examined to establish income inequality order dynamics, while fitting to the context of initial imbalances and short-run dynamics due to unmatched shocks to secondary education (SSE) and higher education (HE). The rejection of the null hypothesis in (Fig 3) is based on the existence of an initial equilibrium, so exploring the following two figures provides insight into the underlying asymmetric legitimacy in Table 4 above. It is clear that income inequality responds negatively and significantly to the expansion of SSE and HE outflows. (Fig 3) clearly shows that the most prominent and overbearing are the progressive SSE and HE shocks. However, the short-run dynamics are characterized by the fact that only positive shocks to SSE and HE reduce income inequality, while negative shocks to SSE and HE have insignificant effects on income inequality. In all cases, short-run inequality adjusts to equilibrium after about six years.

## 5. Conclusion and policy recommendation

Income inequality has widened in both developed and developing countries over the past decade.

Although China is one of the fastest-growing economies in the world, it continues to challenge concerns about rising income inequality as growth is unevenly distributed across different segments of society. The World Inequality Lab found that the richest 10 percent of China's population own nearly 70 percent of total household wealth. Research carried out at the onset of the pandemic has shown that covid restrictions are eating into rural wages. According to a 2020 rural survey by Stanford University researchers, nearly three-quarters of respondents said people who typically work in the field have been unable to do so because of the pandemic. More than 90% said covid controls had reduced their income, and such restrictions on movement have returned in the past six months. This paints a dire picture of inequality, making the study of income distribution an important topic for researchers and policymakers. This study provides evidence that human capital, as measured by educational attainment, plays a key role in reducing income inequality. A fairer distribution of education makes a significant

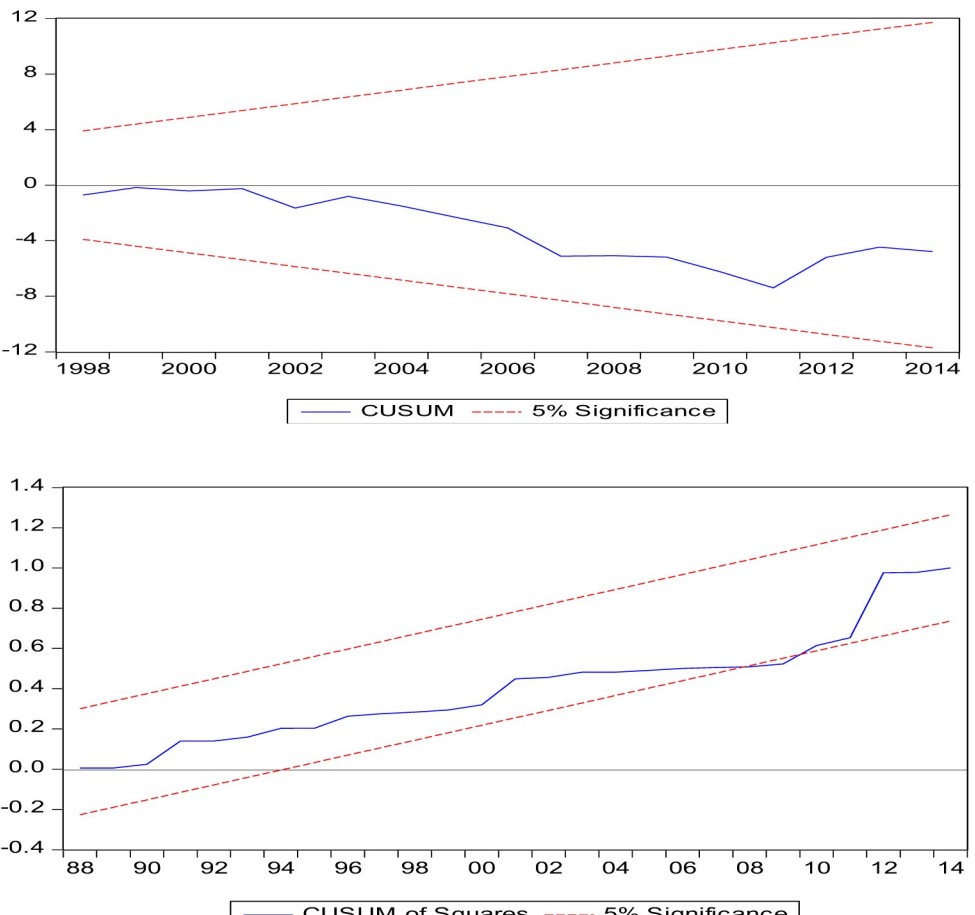

**Fig 1. CUSUM and CUSUMSQ tests for Model-1.**

contribution to reducing income inequality, based on the fact that higher educational attainment reduces educational inequality and thus contributes to reducing income inequality.

Using annual data from 1975 to 2020, this study empirically reveals the link between human capital measured by educational attainment educational inequality and income inequality in the Chinese context. The nonlinear and asymmetric ARDL cointegration approach employed in the current study supports and explores the possibility of nonlinear and asymmetric relationships. The empirical results support the existence of an asymmetric co-integration relationship among secondary education level, higher education level, educational inequality, income inequality, physical capital and economic growth. The long-term asymmetric parameters of the NARDL model reflect that the positive shocks of secondary school education (SSE) and higher education (HE) are negatively correlated with the income Gini coefficient. The adverse shocks of SSE and HE stimulate the Gini coefficient of income, but the effect of SSE on the Gini coefficient of income is not significant, while that of HE is significant. This study also considered the relationship between changes in the income Gini coefficient (income inequality) and education Gini coefficient (educational inequality), showing that there is a positive relationship between changes in income inequality and educational inequality. The impact of economic growth on the Gini coefficient of income is significantly positive in the long run, thus exacerbating income inequality. Furthermore, the stock of physical capital has a significant adverse effect on income inequality in the long run. The second model based

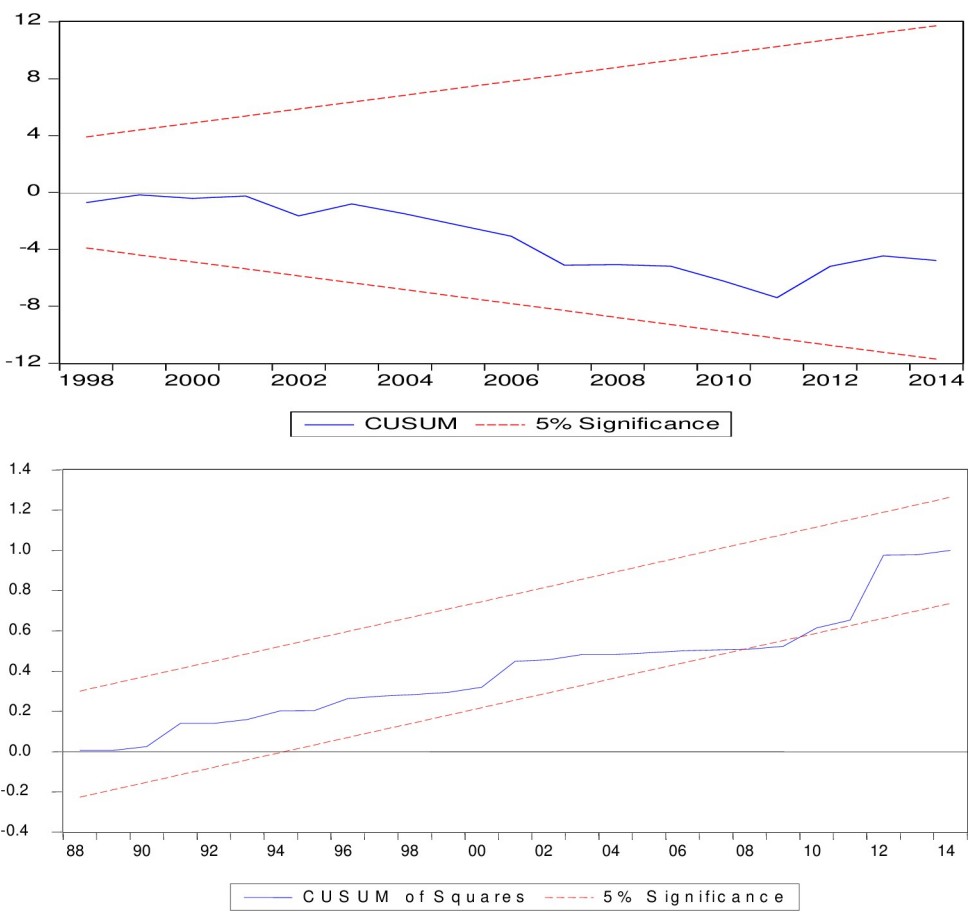

**Fig 2. CUSUM and CUSUMSQ tests for Model-2.**

on the Gini coefficient of education (educational inequality) shows that secondary education has a significant negative impact on the Gini coefficient of education, while the square of secondary education has a significant positive impact on the Gini coefficient of education. This relationship between educational inequality and secondary educational attainment clearly validates the U-shaped hypothesis of the Kuznets curve. In addition, higher education significantly promotes educational inequality, while the square of higher education significantly reduces educational inequality, thus verifying the inverted U-shaped Kuznets curve hypothesis between higher education and educational inequality.

The short-term results show that the positive shock of secondary education leads to a more equitable income distribution, while the positive shock of higher education has no significant impact on China's income distribution. Negative shocks to both levels of education do not have a significant impact on income inequality in the short run. In addition, physical capital stock, economic growth, and education Gini coefficients are significantly and positively correlated with income inequality in the short run. In the second model, the short-term effect of education level on the Gini coefficient of education (educational inequality) is quite different from the long-term effect. Both secondary education (SSE) and the square of secondary education ($SSE^2$) have a significant adverse effect on the education Gini coefficient. However, both higher education and the square of higher education contribute significantly to the education Gini coefficient (educational inequality) in the short run.

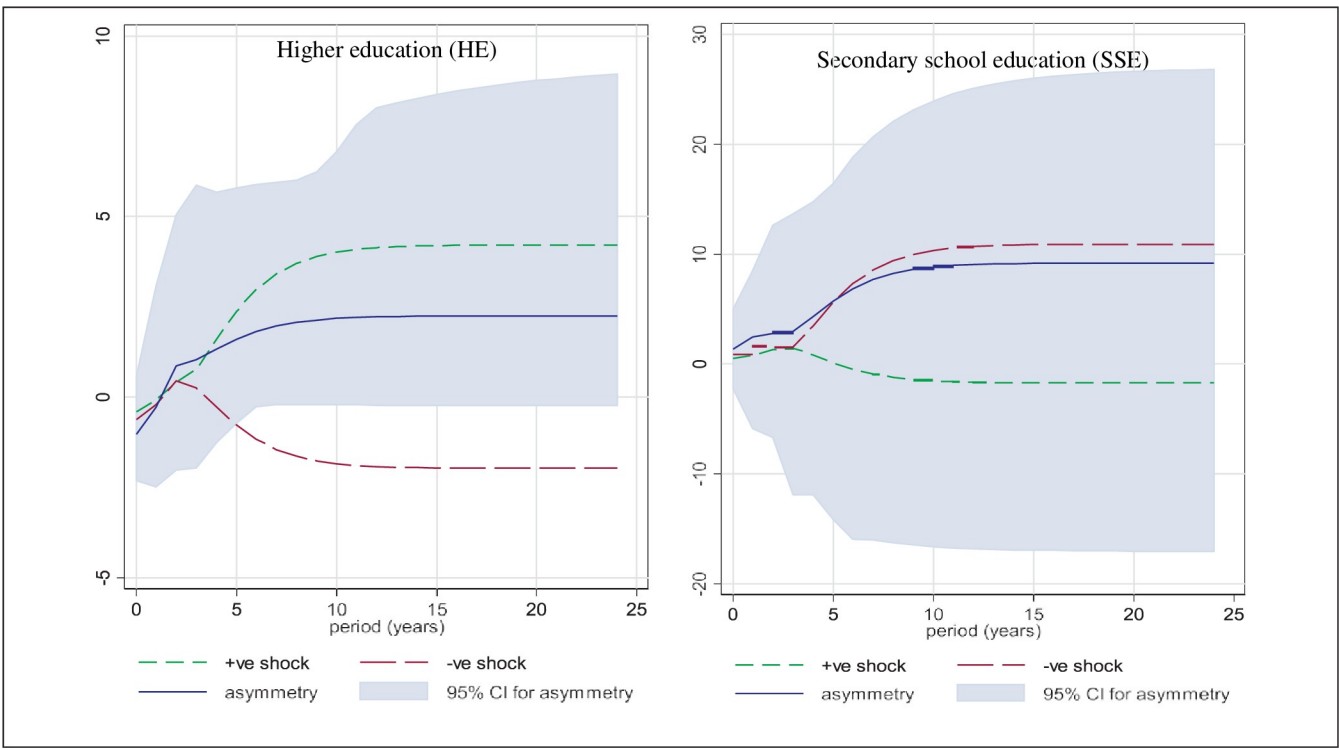

**Fig 3. Dynamic multipliers.** *Note.* 95% bootstrap confidence interval is based on 100 replications.

The empirical results of this study have important implications for China's development policy. First, human capital, as measured by secondary and higher education levels, plays an important role in reducing educational inequality and ultimately income inequality in China. Secondary education remains more effective in the short run in reducing educational inequality, which in turn reduce income inequality. However, long-term income inequality can be reduced by adopting strategies to reduce educational inequality by expanding higher education. Second, the Great Gatsby curve clearly shows that, at a certain point in time, higher income inequality is associated with lower intergenerational mobility, so the important question that arises here is how education affects intergenerational mobility. The distribution (in terms of quantity and quality) of schooling in a population is an important link between income equality and intergenerational mobility. Because income is more unevenly distributed among families, opportunities for economic advancement are distributed even more disproportionately among children. Income sharing and educational attainment among populations is likely to shift from one generation to the next.

The measurement of human capital development based on vocational education and training (VET) and its distribution across the population should also be investigated as a future research direction for reducing income inequality.

## Author Contributions

**Conceptualization:** Jian Chen, Taiming Zhang.

**Data curation:** Mo Xu, Jian Chen.

**Formal analysis:** Mo Xu, Shifeng Chen.

**Investigation:** Shifeng Chen, Jian Chen.

**Methodology:** Mo Xu, Shifeng Chen, Jian Chen.

**Software:** Shifeng Chen, Jian Chen.

**Supervision:** Shifeng Chen.

**Validation:** Taiming Zhang.

**Visualization:** Jian Chen, Taiming Zhang.

**Writing – original draft:** Mo Xu, Jian Chen.

**Writing – review & editing:** Shifeng Chen, Taiming Zhang.

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
