## [Decision Letter · Decision Letter 0]

30 May 2023

PONE-D-23-14637Non-linear relationship between human capital, educational inequality and income inequality, evidence from ChinaPLOS ONE

Dear Dr. Zhang,

Thank you for submitting your manuscript to PLOS ONE. After careful consideration, we feel that it has merit but does not fully meet PLOS ONE’s publication criteria as it currently stands. Therefore, we invite you to submit a revised version of the manuscript that addresses the points raised during the review process. Author needs to response to all Reviewers one by one in detail.Secondly please what is contribution to existing knowledge by this research. 

We look forward to receiving your revised manuscript.

Kind regards,

Muhammad Tayyab Sohail

Academic Editor

PLOS ONE

Journal Requirements:

Reviewers' comments:

Reviewer's Responses to Questions

**Comments to the Author**

1. Is the manuscript technically sound, and do the data support the conclusions?

Reviewer #1: Yes

Reviewer #2: Partly

Reviewer #3: Yes

Reviewer #4: Yes

2. Has the statistical analysis been performed appropriately and rigorously? 

Reviewer #1: Yes

Reviewer #2: Yes

Reviewer #3: Yes

Reviewer #4: Yes

3. Have the authors made all data underlying the findings in their manuscript fully available?

Reviewer #1: Yes

Reviewer #2: Yes

Reviewer #3: Yes

Reviewer #4: Yes

4. Is the manuscript presented in an intelligible fashion and written in standard English?

Reviewer #1: No

Reviewer #2: Yes

Reviewer #3: Yes

Reviewer #4: Yes

5. Review Comments to the Author

Reviewer #1: Reviewer

EVALUATION

Carefully reviewed the manuscript "The Nonlinear Relationship Between Human Capital, Educational Inequality, and Income Inequality: Evidence from China." The topic is interesting, but before your work will be recommended or will be given any possible acceptance; comments must be incorporated for improving the quality of your work.

1. The “title” of the study should be descriptive, and concise, and others should easily understand.

2. In the "Abstract", the description of the research purpose, method, and data time period should be explained in one sentence. Please avoid abbreviations for methods (NARDL) or variables (HE, SSE), instead write full explanations.

3. The second paragraph of the Introduction clarifies the measurement of human capital, but need to cite such evidences. The author needs to focus more on the research problem and the policy-level contribution. Also, the authors should explain the research contributions more explicitly.

4. The literature section should include more up-to-date literature specifically addressing the links between human capital and income inequality, and between educational inequality and income inequality. The authors indicate a literature gap at the end of the literature section, but it needs to be explained more explicitly.

4. The interpretation of the results is insufficient. Please make a more critical comparison of the findings with those in the literature. Methods of estimations and study findings should be clear, detached, and stand-alone. All tables must be in a Word file, formatted in A-4 size with at least 12-point size font.

7. “Conclusion” reiterates the results. It is completely undesirable. The results should be summarized in the first three sentences of the Conclusion. There are practically no policy implications, study limitations, and future research suggestions. Please take note!

8. Ensure that your manuscript is well edited for English language and technical expressions

Reviewer #2: Comments on the revised manuscript: Non-linear relationship between human capital, educational inequality and income inequality, evidence from China.

Dear Dr. Muhammad Tayyab Sohail

Academic Editor

PLOS ONE

Thank you for giving me the opportunity to read and review this manuscript. I have the following comments:

- It is necessary to explain the justifications that were invoked when defining the model variables. By referring to the theoretical framework that confuses these variables, or by referring to the empirical literature in this context. Therefore, the authors should give clear justifications for choosing the study variables.

- The study uses the NARDL model to capture the nonlinear relationship with NARDL (Shin et al., 2014). Although using NARDL developed by (Shin et al., 2014) is a suitable technique for capturing the asymmetric effects of independent variables on a specific dependent variable, however, I think that this technique is more suitable if independent variables are characterized by volatility (increase and decrease), for example, oil prices or financial data. But this is not realized in the case of this study. This model requires observable volatility in the independent variables. The authors need to strengthen their arguments for choosing this model.

- On the other hand, there are many developments since the traditional linear ARDL model of (Pesaran et al., 2001) to capture asymmetric cointegration such as the QARDL model provided by (Cho et al., 2015) or dynamic ARDL (Jordan & Philips, 2018). Therefore, I think that the methodological choice should be better defended.

- This study uses annual time series data from 1975 to 2020. I believe that during this period structural changes occurred in the Chinese economy, which naturally affect the behavior of the variables included in the study. Thus, it is appropriate to test the stability of the time series using some tests that take structural breaks into account to determine the degree of integration of the time series more accurately (e.g. Clemente et al., 1998; Lee & Strazicich, 2003)

General comment: I feel that considering the above notes will improve the manuscript to be more rigorous.

Cho, J. S., Kim, T., & Shin, Y. (2015). Quantile cointegration in the autoregressive distributed-lag modeling framework. Journal of Econometrics, 188(1), 281–300. https://doi.org/https://doi.org/10.1016/j.jeconom.2015.05.003

Clemente, J., Montañés, A., & Reyes, M. (1998). Testing for a unit root in variables with a double change in the mean. Economics Letters, 59(2), 175–182.

Jordan, S., & Philips, A. Q. (2018). Cointegration testing and dynamic simulations of autoregressive distributed lag models. The Stata Journal, 18(4), 902–923.

Lee, J., & Strazicich, M. C. (2003). Minimum Lagrange multiplier unit root test with two structural breaks. Review of Economics and Statistics, 85(4), 1082–1089.

Pesaran, M. H., Shin, Y., & Smith, R. J. (2001). Bounds testing approaches to the analysis of level relationships. Journal of Applied Econometrics, 16(3), 289–326.

Shin, Y., Yu, B., & Greenwood-Nimmo, M. (2014). Modelling asymmetric cointegration and dynamic multipliers in a nonlinear ARDL framework. In Festschrift in honor of Peter Schmidt (pp. 281–314). Springer.

Reviewer #3: This study investigates the relationship between human capital and income inequality for China using a nonlinear asymmetric framework. Long-run and short-run asymmetries over the period 1975-2020 are estimated with the help of NARDLmodel. The theoretical basis of the study is well established, policy recommendations are well presented. However, the following requests should be performed:

1. In the introduction, the next parts of the study should be explained.

2. What are the underlying reasons for determining the 1975-2020 period?

3. It seems appropriate to compare the long-term findings with other similar empirical studies.

4. The main limitations of the study and recommendations for future studies should be presented in detail in the conclusion.

5. The following studies should be included in the literature:

a) https://doi.org/10.1007/s11205-021-02641-7

b) https://dergipark.org.tr/en/download/article-file/470390

c) https://acikerisim.nku.edu.tr/xmlui/handle/20.500.11776/7682

Reviewer #4: Dear Authors,

It is an interesting topic. However, there are some drawnbacks you should address:

1. Please clearly emphasize into Introduction, which other studies addressed this topic, which is the novelty of your research in terms of variables, methodology or findings:

2. Please support the coice for NARDL with previous studies for such topics.

3. Please correlate all your results with findings of previous studies.

4. Extend your policy recommendations and make them more specific.

5. Elaborate some limitations and expand directions for future research cause this topic can be developed greatly.

6. Please fix some typos.

6. PLOS authors have the option to publish the peer review history of their article (what does this mean?). If published, this will include your full peer review and any attached files.

Reviewer #1: **Yes: **Dr Arshad Ali

Reviewer #2: **Yes: **Ibrahim Mohamed Ali Ali

Reviewer #3: No

Reviewer #4: No

---

## [Author Response · Author response to Decision Letter 0]

14 Jun 2023

Response to Reviewers

Dear Editor-in-Chief  

PONE-D-23-14637

Non-linear relationship between human capital, educational inequality and income inequality, evidence from China

PLOS ONE

Thank you for giving us the opportunity to submit a revised draft of the manuscript “Non-linear relationship between human capital, educational inequality and income inequality, evidence from China” for publication in PLOS ONE. We appreciate the time and effort that you and the reviewers dedicated to providing feedback on our manuscript and are grateful for the insightful comments on and valuable improvements to our paper. We have incorporated all the suggestions made by the reviewers. Please see below, for a point-by-point response to the reviewers’ comments and concerns. 

Reviewer 1:

1. The “title” of the study should be descriptive, and concise, and others should easily understand.

Response to Reviewer: The manuscript title has been corrected accordingly.

2. In the "Abstract", the description of the research purpose, method, and data time period should be explained in one sentence. Please avoid abbreviations for methods (NARDL) or variables (HE, SSE), instead write full explanations.

Response to Reviewer: The reviewer is correct; we have highlighted the required correction in yellow in the Abstract section.

3. The second paragraph of the Introduction clarifies the measurement of human capital, but need to cite such evidences. The author needs to focus more on the research problem and the policy-level contribution. Also, the authors should explain the research contributions more explicitly.

Response to Reviewer: We have made the necessary corrections accordingly.

4. The literature section should include more up-to-date literature specifically addressing the links between human capital and income inequality, and between educational inequality and income inequality. The authors indicate a literature gap at the end of the literature section, but it needs to be explained more explicitly.

Response to Reviewer: The latest literature included in the literature section.

4. The interpretation of the results is insufficient. Please make a more critical comparison of the findings with those in the literature. Methods of estimations and study findings should be clear, detached, and stand-alone. All tables must be in a Word file, formatted in A-4 size with at least 12-point size font.

Response to reviewer: We have incorporated all corrections accordingly.

7. “Conclusion” reiterates the results. It is completely undesirable. The results should be summarized in the first three sentences of the Conclusion. There are practically no policy implications, study limitations, and future research suggestions. Please take note!

Response to Reviewer: We have incorporated all corrections accordingly.

8. Ensure that your manuscript is well edited for English language and technical expressions.

Response to Reviewer: The language of the manuscript is improved.

Reviewer 2:

- It is necessary to explain the justifications that were invoked when defining the model variables. By referring to the theoretical framework that confuses these variables, or by referring to the empirical literature in this context. Therefore, the authors should give clear justifications for choosing the study variables.

Response to Reviewer: The plausibility of the variables in the model has been clearly established.

- The study uses the NARDL model to capture the nonlinear relationship with NARDL (Shin et al., 2014). Although using NARDL developed by (Shin et al., 2014) is a suitable technique for capturing the asymmetric effects of independent variables on a specific dependent variable, however, I think that this technique is more suitable if independent variables are characterized by volatility (increase and decrease), for example, oil prices or financial data. But this is not realized in the case of this study. This model requires observable volatility in the independent variables. The authors need to strengthen their arguments for choosing this model.

Response to Reviewer: The arguments over the choice of variables in models and methods has intensified (highlighted in yellow).

- On the other hand, there are many developments since the traditional linear ARDL model of (Pesaran et al., 2001) to capture asymmetric cointegration such as the QARDL model provided by (Cho et al., 2015) or dynamic ARDL (Jordan & Philips, 2018). Therefore, I think that the methodological choice should be better defended.

Response to Reviewer: The reviewer is right, the choice of methodology has been vigorously defended.

- This study uses annual time series data from 1975 to 2020. I believe that during this period structural changes occurred in the Chinese economy, which naturally affect the behavior of the variables included in the study. Thus, it is appropriate to test the stability of the time series using some tests that take structural breaks into account to determine the degree of integration of the time series more accurately (e.g. Clemente et al., 1998; Lee & Strazicich, 2003).

Response to Reviewer: Clement et al. (1998) test has been applied to detect structural breaks, as highlighted in Table 2.

Reviewer 3:

1. In the introduction, the next parts of the study should be explained.

Response to Reviewer: The Introduction section has been improved accordingly.

2. What are the underlying reasons for determining the 1975-2020 period?

Response to Reviewer: The main reasons for selecting time periods for the data have been identified as highlighted in yellow.

3. It seems appropriate to compare the long-term findings with other similar empirical studies.

Response to Reviewer: Yes, the reviewer is right, we have compared the long-term results with related studies.

4. The main limitations of the study and recommendations for future studies should be presented in detail in the conclusion.

Response to reviewer: The limitation of the study has been identified.

5. The following studies should be included in the literature:

a) https://doi.org/10.1007/s11205-021-02641-7

b) https://dergipark.org.tr/en/download/article-file/470390

c) https://acikerisim.nku.edu.tr/xmlui/handle/20.500.11776/7682

Response to Reviewer: The above important relevant studies have been included in the literature.

Reviewer #4: 

It is an interesting topic. However, there are some drawnbacks you should address:

1. Please clearly emphasize into Introduction, which other studies addressed this topic, which is the novelty of your research in terms of variables, methodology or findings:

Response to Reviewer: The novelty of this study has been explicitly discussed.

2. Please support the coice for NARDL with previous studies for such topics.

Response to Reviewer: The reviewer is right, the choice of methodology has been vigorously defended.

3. Please correlate all your results with findings of previous studies.

Response to reviewer: Yes, the reviewer is right, we have compared the long-term results with previous related studies.

4. Extend your policy recommendations and make them more specific.

Response to Reviewer: Policy recommendations are more clearly explained.

5. Elaborate some limitations and expand directions for future research cause this topic can be developed greatly.

Response to Reviewer: Limitations of the study has been identified.

6. Please fix some typos.

Response to Reviewer: Errors are reduced to a certain extent.

---

## [Decision Letter · Decision Letter 1]

10 Jul 2023

Non-linear links between human capital, educational inequality and income inequality, evidence from China

PONE-D-23-14637R1

Dear Dr. Zhang,

We’re pleased to inform you that your manuscript has been judged scientifically suitable for publication and will be formally accepted for publication once it meets all outstanding technical requirements.

Kind regards,

Muhammad Tayyab Sohail

Academic Editor

PLOS ONE

Additional Editor Comments (optional):

Reviewers' comments:

Reviewer's Responses to Questions

**Comments to the Author**

1. If the authors have adequately addressed your comments raised in a previous round of review and you feel that this manuscript is now acceptable for publication, you may indicate that here to bypass the “Comments to the Author” section, enter your conflict of interest statement in the “Confidential to Editor” section, and submit your "Accept" recommendation.

Reviewer #1: All comments have been addressed

Reviewer #4: All comments have been addressed

2. Is the manuscript technically sound, and do the data support the conclusions?

Reviewer #1: Yes

Reviewer #4: Yes

3. Has the statistical analysis been performed appropriately and rigorously? 

Reviewer #1: Yes

Reviewer #4: Yes

4. Have the authors made all data underlying the findings in their manuscript fully available?

Reviewer #1: Yes

Reviewer #4: Yes

5. Is the manuscript presented in an intelligible fashion and written in standard English?

Reviewer #1: Yes

Reviewer #4: Yes

6. Review Comments to the Author

Reviewer #1: The author incorporated all my comments to improve the quality of manuscript. Thus, I recommend that paper in the current form meet the journal's acceptance requirements.

Reviewer #4: No further comments. My comments were addessed. I think that the paper can be accepted in the current form.

7. PLOS authors have the option to publish the peer review history of their article (what does this mean?). If published, this will include your full peer review and any attached files.

Reviewer #1: **Yes: **Dr Arshad Ali

Reviewer #4: No

---

## [Editor Report · Acceptance letter]

24 Jul 2023

PONE-D-23-14637R1 

Non-linear links between human capital, educational inequality and income inequality, evidence from China 

Dear Dr. Zhang:

I'm pleased to inform you that your manuscript has been deemed suitable for publication in PLOS ONE. Congratulations! Your manuscript is now with our production department. 

Kind regards, 

on behalf of

Dr. Muhammad Tayyab Sohail 

Academic Editor

PLOS ONE